# Problem-Based Learning in Türkiye: A Systematic Literature Review of Research in Science Education

**Behiye Akcay [1,*] and İbrahim Benek [2]**

1   Department of Science Education, İstanbul University-Cerrahpaşa, Istanbul 34500, Türkiye
2   Edremit Science and Art Centre, Ministry of National Education, Van 65040, Türkiye;
    ibrahimbenek11@gmail.com
*   Correspondence: behiye.akcay@iuc.edu.tr

**Abstract:** This study aimed to conduct a systematic literature review of research to provide an overview of the key findings and trends in studies on problem-based learning within the context of science education in Türkiye. To achieve this goal, descriptive content analysis was used in this study. Articles and graduate theses conducted in Türkiye between 2000 and December 2023 were included in this study. The Turkish Academic Network and Information Center (TR Dizin) and National Thesis Center databases were used to access the articles and theses. The purposive sampling method known as the criterion sampling method was employed in this study, resulting in the inclusion of 133 studies, including 37 articles and 96 graduate theses. To facilitate data analysis, we developed a coding form. The results of this study showed that PBL had a positive impact on 34 different skills, and it had no impact on 11 different skills. Across all reviewed studies, the most preferred research design was the quasi-experimental design. There was limited inclusion of final-year students in the samples at various school levels, and researchers mainly preferred physics subjects for their studies.

**Keywords:** problem-based learning; science education; Türkiye

## 1. Introduction

Problem-based learning (PBL) is a widely recognized pedagogical approach that has garnered substantial attention in education worldwide. Originating from medical education in the late 1960s, PBL has since evolved and been implemented across various educational contexts, including higher-education institutions [1]. According to Savery and Duffy [2], PBL is grounded in several key principles that shape its implementation, including using authentic, ill-structured problems as a starting point for learning, active engagement of students in problem-solving activities, collaborative learning within small groups, facilitator guidance rather than traditional teaching, and self-directed learning and reflection [2].

PBL has gained significant recognition and relevance in science education due to its effectiveness in promoting a deeper understanding of scientific concepts, enhancing students' problem-solving abilities, and fostering students' critical thinking skills as they analyze, evaluate, and synthesize information to solve problems [3]. Additionally, it helps prepare students for future careers in science-related fields by equipping them with skills, such as critical thinking, problem solving, teamwork, and adaptability, which are highly valued by employers. It is characterized by its student-centered, inquiry-driven, and collaborative nature. Students engage in a dynamic and collaborative inquiry process to explore and solve complex, authentic problems [4].

This approach emphasizes active learning, critical thinking, problem-solving skills, and the application of knowledge in practical contexts [5]. PBL engages students actively in the learning process, making science more interesting and relevant. They are motivated to seek out information, conduct research, and interact with peers to find solutions [6]. This hands-on approach encourages a deeper understanding of scientific concepts. PBL shifts

the role of the teacher from a lecturer to a facilitator. Students take responsibility for their own learning, fostering self-directed learning skills and a sense of ownership over their education. The teacher guides and supports the learning process, providing resources and feedback when needed [7].

PBL typically begins with the presentation of scientific concepts and principles within the context of real-world problems or scenarios. It should be authentic, ill-structured, complex, and relevant to the content being studied in order to facilitate students to reach their own conclusions by doing open inquiry [8]. This serves as a trigger for learning. Students take ownership of their learning: they decide what information they need to acquire, how to acquire it, and how to apply it to solve the problem [6]. This contextualization helps students see the relevance of what they are learning and how it can be applied in practical situations. Students often work in groups, requiring effective teamwork, communication, and collaboration skills, essential in scientific research and professional settings. PBL often involves a cycle of research, discussion, and reflection. It encourages students to explore and understand the problem; identify relevant concepts and principles; generate hypotheses and solutions; and construct arguments to support their solutions [9]. Also, students are encouraged to reflect on their learning process, their understanding of the problem, and their solutions [10].

## 2. Theoretical Foundations

PBL is a pedagogical approach rooted in several well-established educational theories, each contributing to its effectiveness in science education. The theoretical underpinnings of PBL in science education, including constructivism, cognitive load theory, and situated learning, provide a strong foundation for understanding its effectiveness [1,2,10–12].

Constructivist learning theory posits that learners actively build their own understanding and knowledge through a process of constructing mental models based on their experiences and prior knowledge [13]. In the context of PBL in science education, this theory aligns with the idea that students construct scientific knowledge by engaging with authentic problems. In a PBL setting, students are presented with complex, real-world scientific problems that require them to draw upon their existing knowledge, engage in research, collaborate with peers, and synthesize information to generate solutions [2,12,14]. According to Leite, Dourado, and Morgado, teamwork and cooperative learning provide very effective learning environments for PBL [15]. This active engagement allows students to construct a deeper and more meaningful understanding of scientific concepts. By grappling with problems, they refine their mental models of scientific phenomena and develop problem-solving skills critical to scientific inquiry [2].

Cognitive load theory (CLT) explores how the cognitive load imposed on learners can impact their ability to process information and learn effectively [11,12,16]. PBL, when designed well, aligns with CLT principles by managing cognitive load in science education. In a PBL environment, the cognitive load is distributed over time, allowing students to build their understanding progressively. They start with a problem and incrementally acquire the necessary knowledge and skills to solve it. This "scaffolding" approach minimizes cognitive overload, ensuring that students can focus on meaningful learning experiences [17,18].

Situated learning theory [12,19,20] emphasizes the importance of learning in context. It argues that learning is most effective when it occurs within authentic, real-world situations. PBL in science education aligns with this theory by placing scientific knowledge and skills in relevant contexts. In PBL scenarios, students are immersed in situations that resemble the complexities and challenges faced by scientists in their research or practical work. This contextualization not only enhances the relevance of scientific content but also fosters the development of problem-solving skills that are transferable to real-world scientific endeavors [10].

These theories support the idea that PBL promotes active engagement, aligns with the cognitive processes of learners, and fosters the development of skills necessary for scientific inquiry. By integrating these theories into the design and implementation of PBL, educators

can create a rich learning environment that enhances both scientific knowledge acquisition and problem-solving abilities among students [11].

## 3. Implementation of PBL in Türkiye

PBL was introduced to Turkish education by establishing medical faculties that adopted the curriculum of Maastricht University in the early 2000s [21]. Since then, PBL has gradually expanded to other disciplines and levels of education, including K-12, undergraduate, and graduate programs. PBL's implementation in Türkiye has been characterized by adapting international models to suit the local context [22].

The introduction of PBL in Turkish education has faced several challenges. Although the Turkish Ministry of National Education (MoNE) has restructured the curriculum according to constructivism since 2004, one of the main significant challenges is the traditional lecture-based teaching approach deeply ingrained in the educational system [23]. Resistance to change among educators and students has been a barrier to the widespread adoption of PBL. Language has also been challenging, particularly in disciplines with a substantial theoretical component. PBL often requires students to work in groups and communicate effectively in the target language, which may be a foreign language in some cases [21]. Another challenge is the need for well-trained faculty members who can facilitate PBL effectively. Training faculty members to transition from a didactic teaching style to a facilitative one that supports student-centered learning has been an ongoing process [23].

Despite these challenges, research in Türkiye has highlighted several benefits associated with PBL implementation. Studies have reported that PBL enhances students' critical thinking, problem-solving skills, and motivation for learning [24]. PBL has also been found to promote teamwork, communication skills, and a deeper understanding of the subject matter [21]. Furthermore, research suggests that PBL can contribute to developing lifelong learning skills and better prepare students for the demands of the modern workforce, aligning with Türkiye's goals of improving higher education and workforce readiness [24,25].

In Türkiye, PBL has made significant strides in higher education, with its implementation expanding beyond medical faculties to various disciplines. While challenges related to traditional teaching methods and faculty training persist, research indicates that PBL offers many benefits to students, including improved critical thinking, problem-solving skills, and subject matter comprehension [26]. As Türkiye continues to adapt and refine its approach to PBL, the potential for this student-centered pedagogy to positively impact Turkish education remains promising. Developing successful PBL should start with aligning PBL activities with the curriculum standards and learning objectives specified by the Turkish Ministry of National Education, ensuring that PBL projects address the key science concepts and skills that students are expected to learn. In accordance with the 2018 Primary Education Science Program issued by the Ministry of National Education, it is imperative to select science topics or problems that are both age-appropriate and relevant to students' daily lives. This approach is crucial for igniting curiosity, sustaining interest, and establishing connections to their own experiences, thereby facilitating meaningful learning [27].

In recent years, PBL has gained prominence in Türkiye as educators and policymakers seek innovative and effective teaching methods to enhance students' critical thinking, problem-solving abilities, and overall learning experiences. Türkiye, situated at the crossroads of Europe and Asia, boasts a diverse educational landscape that spans from primary schools to universities. As Turkish educators grapple with the challenges of preparing students for an increasingly complex and dynamic global environment, the adoption and implementation of PBL have emerged as a potential solution. This educational approach aligns with the broader goals of fostering autonomous learners equipped to navigate real-world challenges effectively [25].

The importance of conducting a descriptive content analysis of PBL studies in Türkiye becomes evident when considering the need to map the current state of PBL research,

identify trends, and understand the contextual factors influencing its implementation. Such an analysis can offer valuable insights into the evolution of PBL within the Turkish educational system and highlight areas where further research and development may be necessary. This study aims to provide a comprehensive overview of the existing body of literature on PBL in Türkiye. Within the scope of this study, answers were sought to the following research questions:

1. What is the distribution of studies according to publication type?
2. What is the distribution of studies according to publication year?
3. What is the distribution of studies according to different research variables?
4. What is the distribution of studies according to the research methods used?
5. What is the distribution of studies according to level of education?
6. What is the sample distribution in the studies according to grade level?
7. What is the distribution of topics addressed in studies?
8. What is the distribution of studies according to the duration of the research?
9. What is the distribution of the results obtained from the studies related to PBL?
10. What is the distribution of studies according to science topics in the studies?

## 4. Materials and Methods

### 4.1. Research Design

In this study, the aim was to conduct a systematic literature review of research in science education related to PBL in science disciplines in Türkiye. To achieve this objective, a qualitative method known as descriptive content analysis was used in this study [28]. The Preferred Reporting Items for Systematic reviews and Meta-Analyses (PRISMA) protocol was used as a guide to conduct this study [29].

In the literature, meta-analysis, meta-synthesis, and descriptive content analysis approaches are used for content analysis. In meta-synthesis studies, which are expressed as a meta-analysis of quality [29,30], only the qualitative dimensions of qualitative or mixed studies conducted in a certain field are evaluated [29]. Meta-synthesis research involves a methodological approach that synthesizes the findings of each area of research included in the study in terms of theory, methods, and data [30]. A meta-analysis, on the other hand, is a method of determining the effect of an independent variable on a dependent variable by calculating the effect size of studies conducted on a particular subject, combining the results of the study and statistical analysis of the obtained research findings [31]. In descriptive content analysis, qualitative and quantitative studies on a particular subject are evaluated and analyzed [32]. Descriptive content analysis is a scientific method in which quantitative and qualitative data are systematically coded, synthesized, and analyzed within the framework of certain themes and patterns [20,32]. A flowchart illustrating the process of identifying pertinent publications for this systematic review is shown in Figure 1.

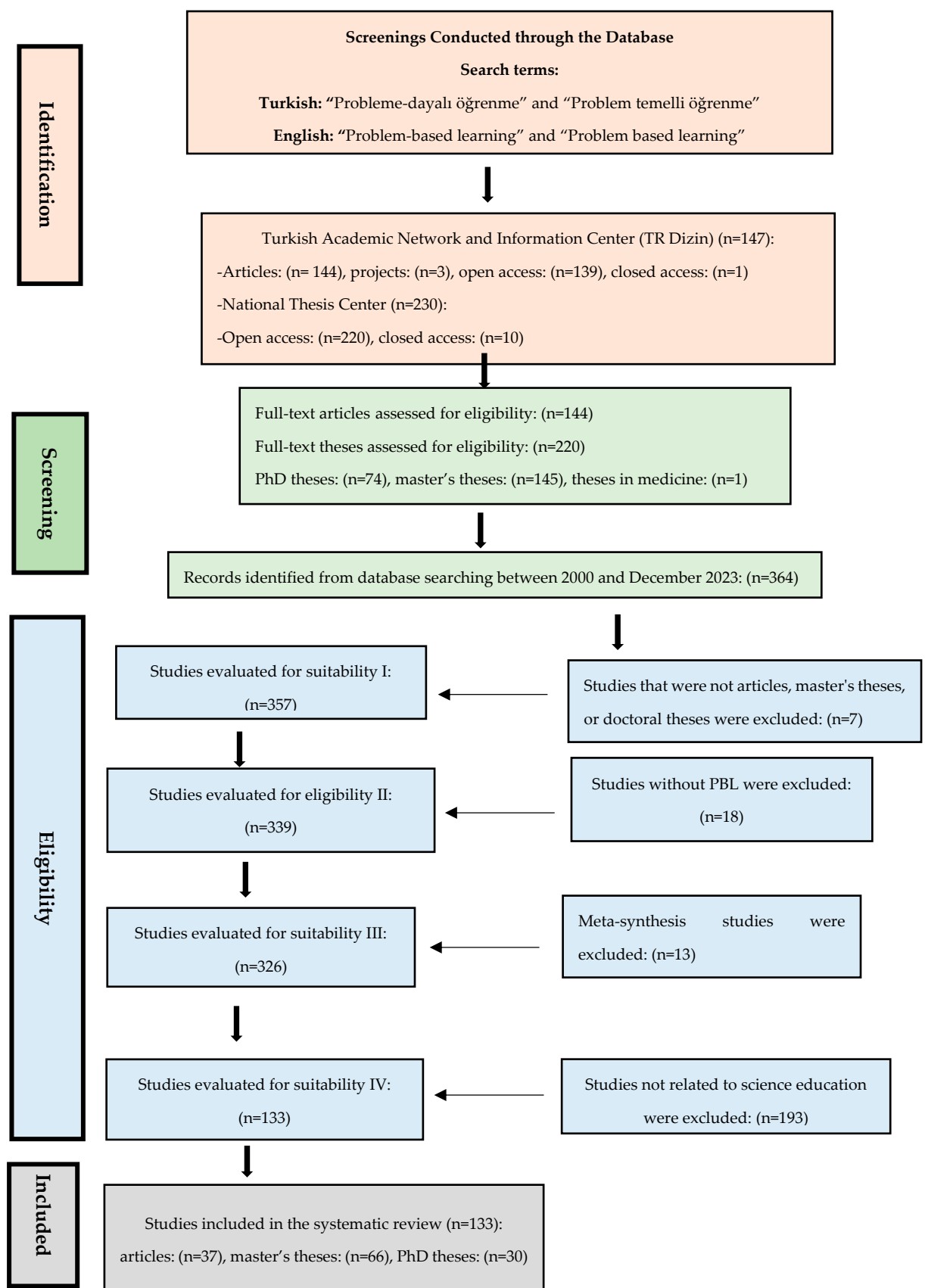

**Figure 1.** Flowchart illustrating the process of identifying pertinent publications for the systematic review.

*4.2. Sample*

This study included a comprehensive search of The Turkish Academic Network and Information Center (TR Dizin) and National Thesis Center databases focusing on PBL in science education in Türkiye between 2000 and 2023. The purposive sampling method known as the criterion sampling method was employed in this study, resulting in the inclusion of a total of 133 studies, including 37 articles, 66 master's theses, and 30 doctoral theses (Appendix A. To identify relevant studies, searches were conducted using keywords, including problem-dayalı öğrenme (problem-based learning), problem temelli öğrenme (problem-based learning) in the search engines of TR Dizin and National Thesis Center databases. These keywords were searched in the titles, abstracts, and keywords of the articles and theses. During the search process, various combinations of these keywords in both Turkish and English were utilized.

Several criteria were considered when selecting articles and graduate theses for inclusion in the research. The selection criteria for the studies included:

- The studies had to be related to the problem-based learning (PBL) approach;
- The studies had to be related to science education;
- The studies had to be published in both Turkish and English;
- The studies had to have been conducted between 2000 and September 2023;
- Articles had to be published in journals indexed in The Turkish Academic Network and Information Center (TR Dizin);
- Graduate theses had to be accessible through the National Thesis Center database;
- The samples in the studies had to consist of pre-service and in-service teachers, including kindergarten, elementary, special education, science, biology, chemistry, and physics teachers;
- The samples in the studies had to consist of kindergarten, elementary, middle school, high school, undergraduate, and graduate students;
- The samples had to be located within the borders of Türkiye;
- The samples had to not be conference papers, projects, books, or reports.

Studies that did not meet the inclusion criteria were excluded from the research. As a result, 133 studies meeting these criteria were identified. These studies were compiled into a single-drive file, and each study was listed with the following information: sequence number, author(s), and publication year to facilitate easy access.

*4.3. Data Collection Tools and Data Analysis*

To facilitate the data analysis in a more accessible and understandable manner, we developed a coding form. Information regarding the articles and theses that met the research criteria was added to this form. The developed form is presented in Table 1.

The Coding Form included information about the author(s)' names, the year of the study, the index of the journal where the study was published, the dependent and independent variables of the study, the research design, the school level and grade level of the study, the qualitative and quantitative data collection tools used in the study, the duration of implementation, the unit/topic addressed in the study, and the results obtained from the study. The Journal Index column was applicable to articles and does not appear in the form of theses. To ensure coding reliability, two independent coders separately reviewed the studies and recorded the results in the coding form. The codings were compared, and the suitability of the codings was determined. The reliability between the coders was calculated as 94%.

**Table 1.** Article/thesis coding form.

| Number | Author(s) | Publication Year | Journal Index | Variables | | Research Design | School Level | Grade | Data Collection Tool | | Content | Results |
|---|---|---|---|---|---|---|---|---|---|---|---|---|
| | | | | Dependent Variable | Independent Variable | | | | Qualitative | Quantitative | | |
| 1 | | | | | | | | | | | | |
| 2 | | | | | | | | | | | | |
| 3 | | | | | | | | | | | | |

## 5. Results

### 5.1. Distribution of Studies According to Publication Type

When looking at the distribution of the studies related to PBL in science disciplines between 2000 and 2023 according to the publication type, it was determined that the studies were mostly conducted at the master's level (49.6%), while the number of studies at the article (27.8%) and doctoral (22.5%) levels was less (Figure 2).

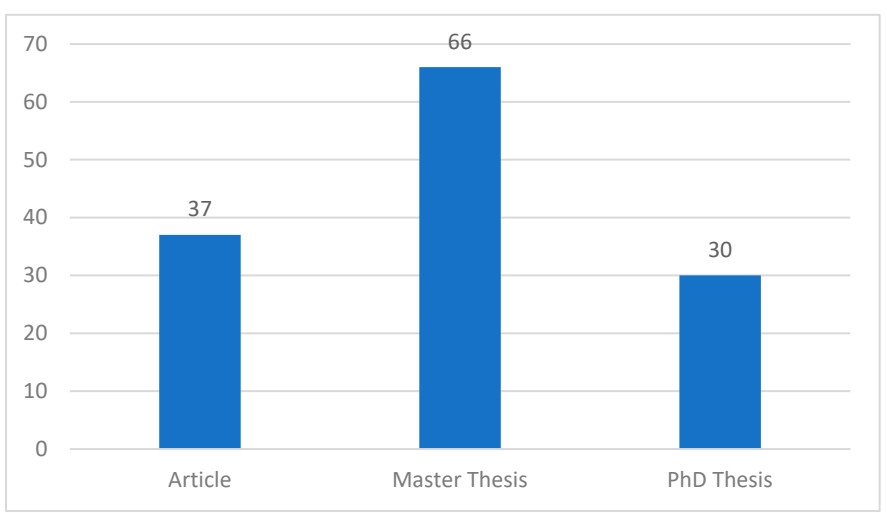

**Figure 2.** Distribution of studies according to publication type.

### 5.2. Distribution of Studies According to Publication Year

When Figure 3 is examined, it can be observed that there were no article studies related to PBL (presumably a specific subject or topic) between 2000 and 2004, or in 2006, 2007, 2011, 2016, and 2022. Similarly, no master's theses were conducted in 2000–2003 or in 2005, and no doctoral theses were conducted in 2000–2002, 2004–2007, and 2022–2023 about PBL. It was also evident that no studies related to PBL were conducted from 2000 to 2003. The first studies related to PBL in Türkiye appear to have started in 2003. It was found that the highest number of articles was produced in 2020 (six articles), the highest number of master's theses appeared in 2019 (seven theses), and the highest number of doctoral theses

appeared in 2012 (five theses) and 2017 (five theses). Additionally, it was observed that the highest number of studies appeared in 2012 (14 studies) and 2017 (13 studies).

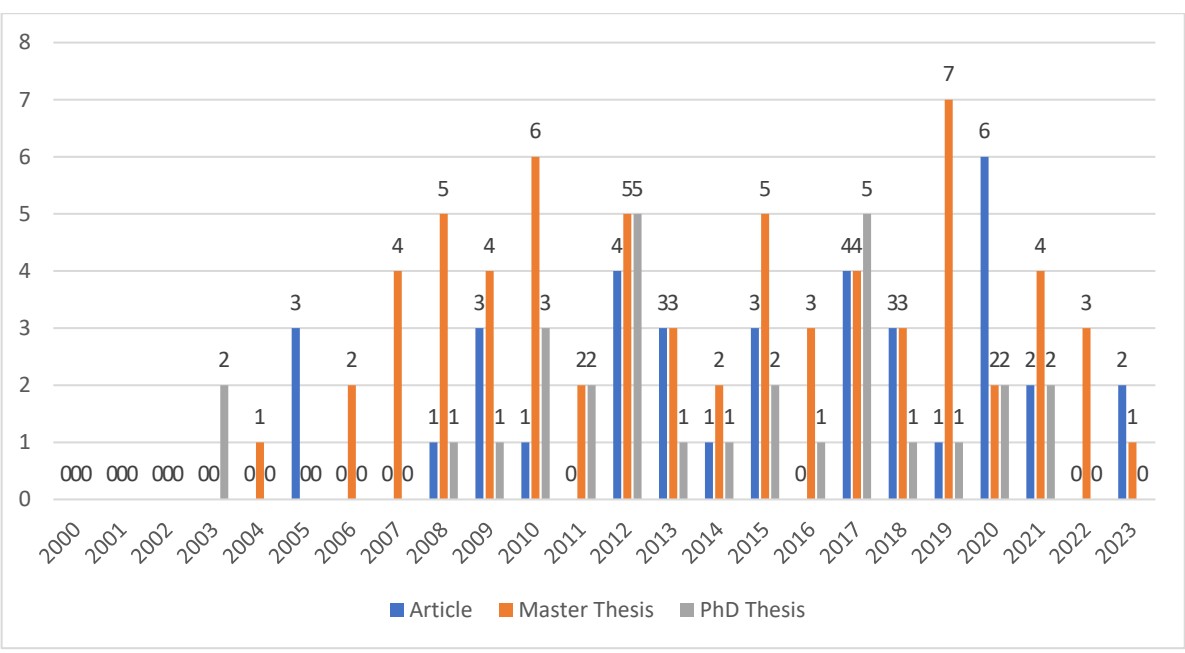

**Figure 3.** Distribution of studies by years.

### 5.3. Distribution of Studies According to Different Research Variables

When we analyzed the studies according to different research variables, it was seen that only two studies were conducted on PBL and these were theoretical/conceptual studies and one measurement tool development study. The remaining 130 studies had all been conducted in the context of examining the impact of PBL methods on different variables. These variables included problem-solving skills, self-efficacy beliefs, academic achievement, misconceptions, opinions, motivation, attitudes, metacognitive awareness, learning strategies, learning style, science process skills, cognitive processing skills, social skills, creative thinking skills, analytical thinking skills, persistence, conceptual understanding, scale development, the development of computer-integrated PBL materials, critical thinking, inquiry-based learning, pedagogical content knowledge (PCK), scientific reasoning ability, module creation and implementation, environmental education, self-directed learning, levels of conceptual structuring, the development and impact assessment of instructional materials, academic risk-taking, scientific literacy, metacognition, energy literacy, career interest, logical thinking, engineering and technology perception, reflective thinking skills for problem solving, digital literacy, reasoning skills, cognitive flexibility, cognitive skill level, the development of fundamental concepts related to heat and temperature, self-regulatory learning skills, and decision-making skills. A total of 69 dependent variables were examined in the studies. The 10 most studied dependent variables with the most impact are presented in Figure 4.

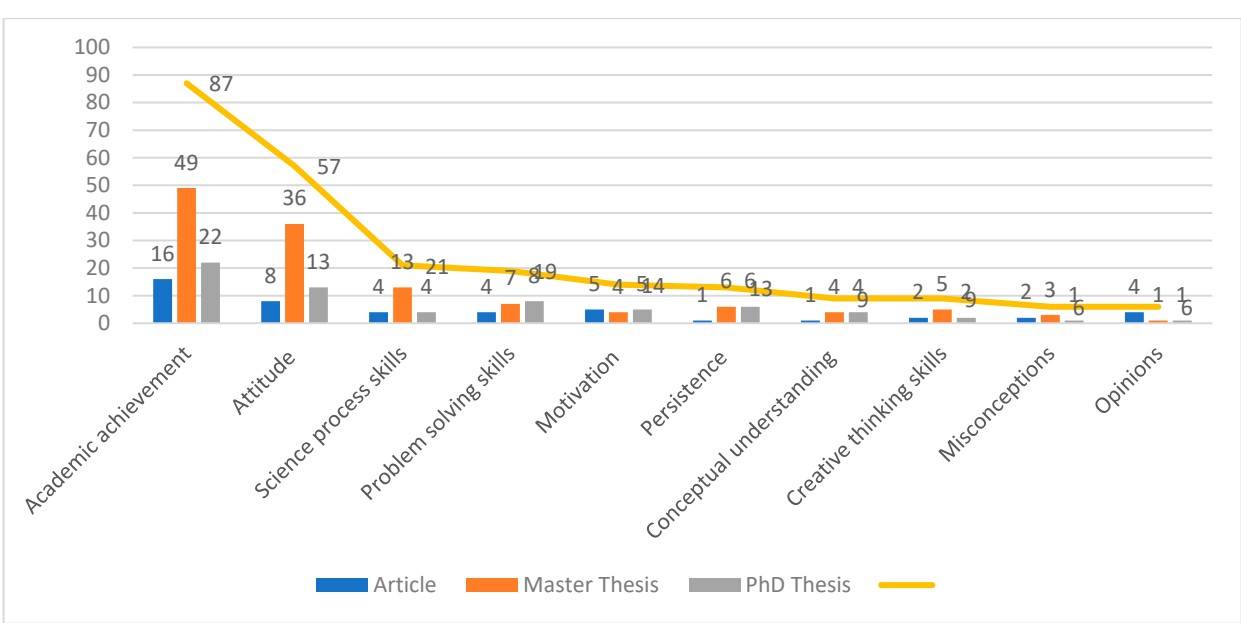

**Figure 4.** The most common dependent variables in the studies.

When Figure 4 is examined, it can be observed that the most frequently studied depended variable in the conducted research was the effect of the method on academic achievement (87). Following this, achievement, attitude (57), science process skills (21), and problem-solving skills (19) were examined, listed in order of their prevalence. Furthermore, when Figure 4 is examined, it is seen that there were fourteen studies on motivation, thirteen studies on persistence, nine studies on conceptual understanding, nine studies on creative thinking skills, six studies on misconceptions, and six studies on student opinions.

### 5.4. Distribution of Studies According to Research Methods Used

Findings regarding the distribution of studies according to the research methods used showed that both quantitative and qualitative methods were used in studies related to PBL. The methods used include "Full Experimental Design", "Semi-Experimental Design", "Weak Experimental Design", "Mixed method", "Case Study", "Action Research", and "Qualitative Research". The methods used in the studies are presented in Figure 5.

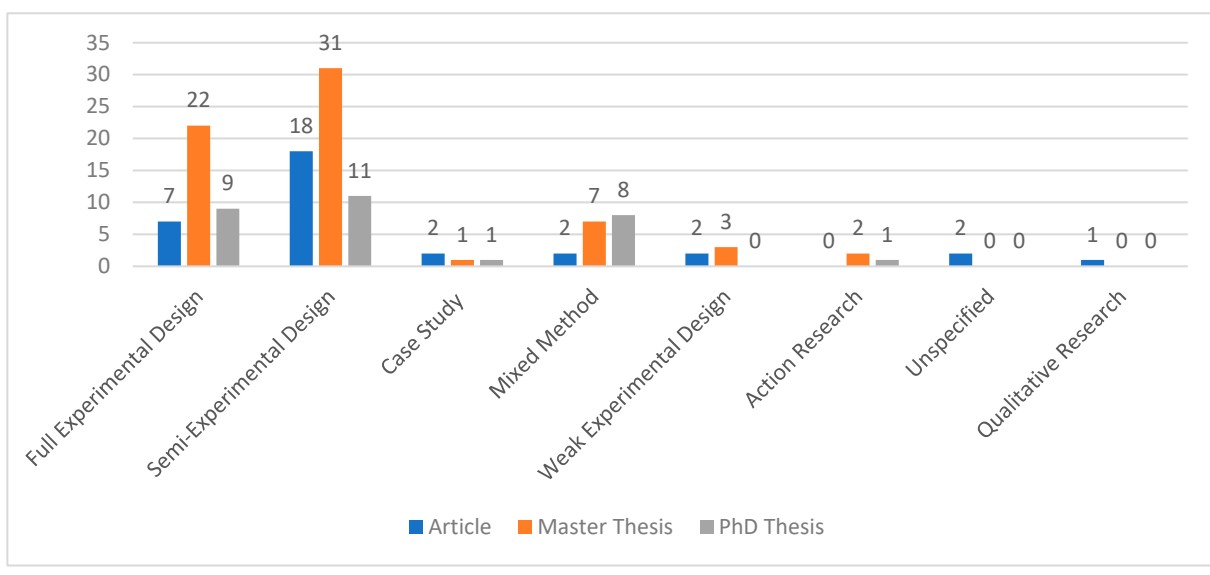

**Figure 5.** Methods used in the studies.

When Figure 5 is examined, it is observed that the most preferred method in these studies related to PBL was the semi-experimental design (60). This method was followed by the full experimental design (38), mixed-method (17), weak experimental design (5), case study (4), action research (3), and qualitative research (1) approaches. In two studies, the method was not clearly specified.

It was determined that the methods used in the studies had changed over the years. The evolution or clustering of the methods used in the studies over the years is presented in Figure 6.

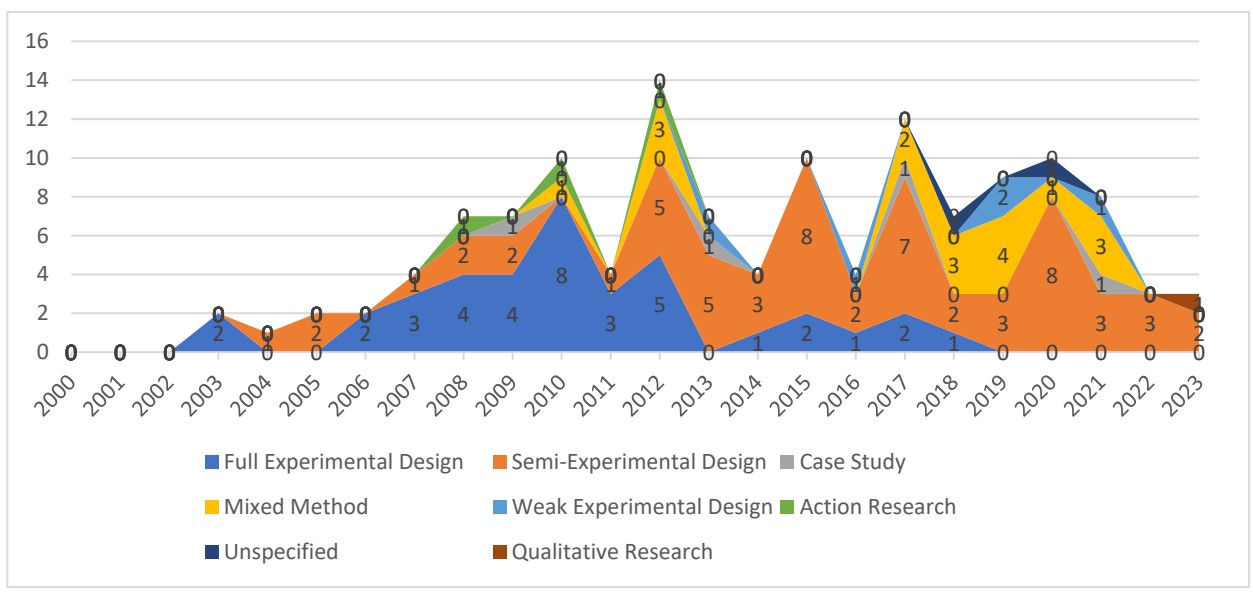

**Figure 6.** Stacked area graph of methods used in the studies.

When examining Figure 5, it is observed that the first studies related to PBL in Türkiye were full experimental studies. It was found that full experimental studies were most common between 2008 and 2012 (24 studies) and that their numbers have gradually decreased in recent years. On the other hand, semi-experimental studies have been on the rise since 2012. Approximately 85% (51) of the semi-experimental studies were conducted between 2012 and 2023. It was determined that the highest number of full experimental studies occurred in 2010 (eight studies), and the highest number of semi-experimental studies appeared in 2015 (eight studies) and 2020 (eight studies). The first qualitative study (action research) was conducted in 2008. It was observed that no mixed-method research was conducted until 2010, four mixed-method studies were conducted between 2010 and 2012, no mixed-method research was conducted between 2013 and 2016, and thirteen mixed-method studies were conducted after 2017. Based on this, it can be concluded that 76.47% (13) of the mixed-method research was conducted in recent years.

*5.5. Distribution of Studies According to Level of Education*

When the distribution of the studies was examined according to the level of education that was studied, it was determined that studies were conducted across all school levels, from kindergarten to university. It was seen that these studies were mostly conducted as master's theses studies (66 studies) (Figure 7).

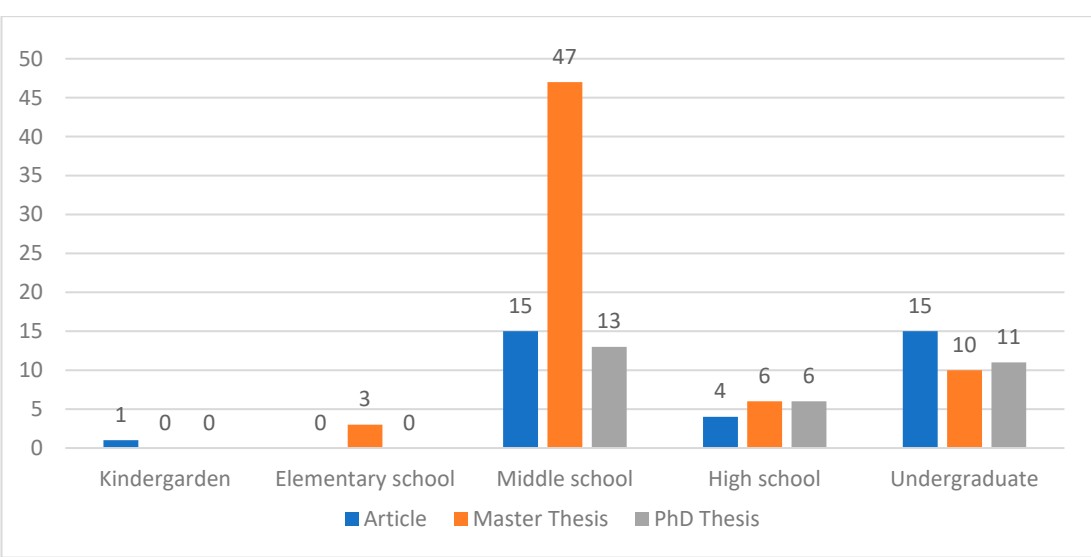

**Figure 7.** Clustered bar chart of research according to the studied level of education.

As shown in Figure 6, it can be observed that there was only 1 study related to PBL at the preschool level, 3 studies at the primary school level, 75 studies at the middle school level, 16 studies at the high school level, and 36 studies at the undergraduate level. Based on this, it can be concluded that more than half of the studies related to PBL are conducted at the middle school level.

*5.6. Distribution of the Samples in the Studies According to Grade Levels*

When the findings regarding the distribution of studies according to grade levels were examined, it was observed that most studies related to PBL were conducted with seventh-grade students. Following that, sixth-grade, fifth-grade, and junior students were the next most commonly selected as samples. In studies conducted at the undergraduate level, primarily junior students were chosen. Another noteworthy result observed when analyzing Figure 7 is the limited or almost non-existent selection of final-year students at the school levels as samples. It was seen that there were eleven studies related to 8th-grade students, three studies related to 4th-grade students, and one study related to senior students, but no studies related to 12th-grade students. Additionally, it is noted that three (3) of the conducted studies were carried out with students diagnosed with special abilities, two (2) undergraduate-level studies did not provide any information about the class level, and in two (2) undergraduate-level studies, more than one class level was selected as the sample. It was concluded that 80% of the studies investigating the effects of problem-based learning were conducted with seventh-grade, sixth-grade, fifth-grade, junior, eighth-grade, or freshman students (Figure 8).

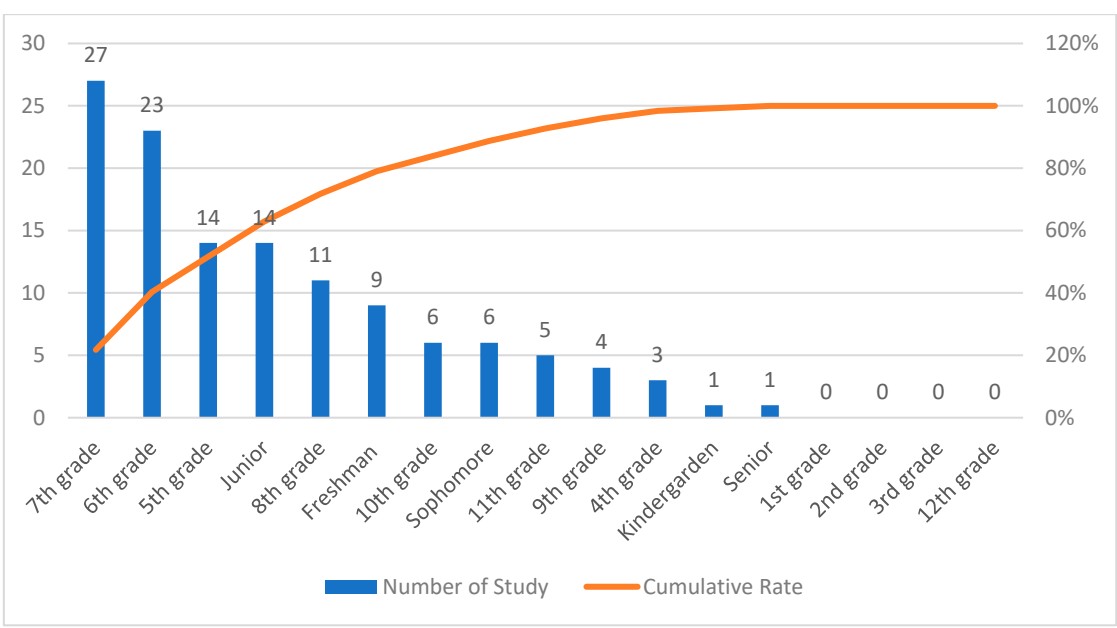

**Figure 8.** Pareto diagram of the samples according to the grade levels studied.

Findings regarding the discipline of the undergraduate samples (Figure 9) used in the research showed that 26 of the studies were conducted with pre-service science teachers (68.42%). Additionally, five studies were related to pre-service chemistry teachers, four studies involved pre-service elementary teachers, two studies involved pre-service physics teachers, and one study used pre-service mathematics teachers. These findings indicated that there were no studies conducted with pre-service biology teachers.

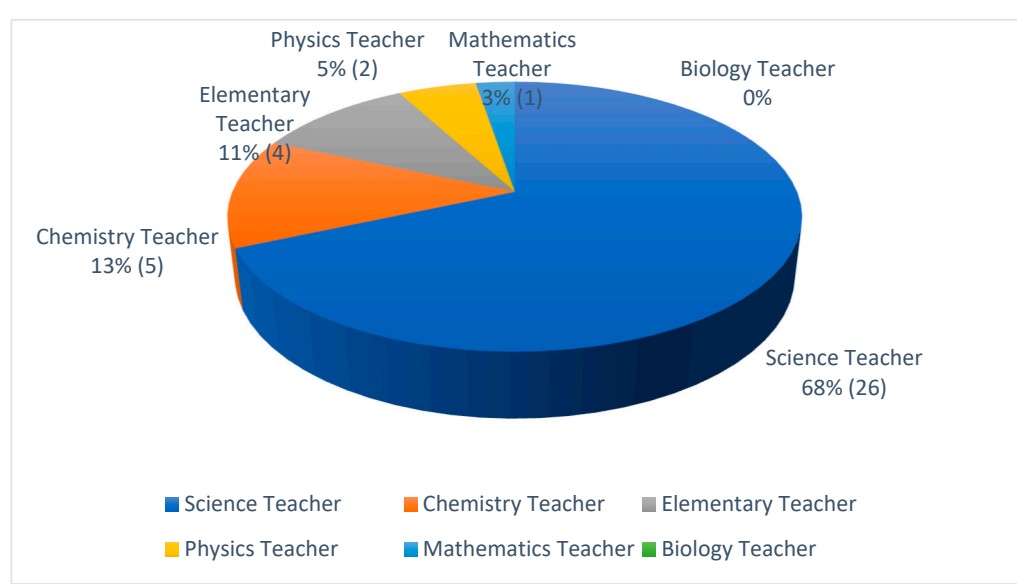

**Figure 9.** Pie chart of research based on the undergraduate samples.

*5.7. Distribution of Topics Addressed in Studies by Subject*

When we look at the distribution of the topics addressed in the studies in Figure 10, we see that the studies primarily focused on physics topics (64). In addition, there were 31 studies related to chemistry topics, 23 studies related to biology topics, 10 studies related to environmental topics, 2 studies related to laboratory science, and 1 study on mathematics topics.

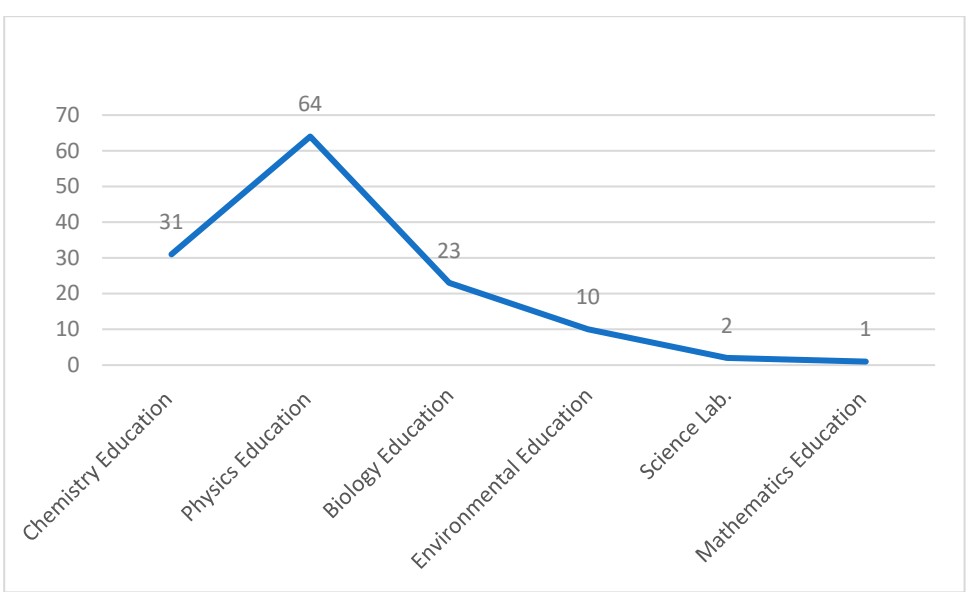

**Figure 10.** Distribution of topics addressed in the studies by subject.

*5.8. Findings Regarding the Duration of the Research*

Upon examining Figure 11, it can be observed that some studies provided information about the duration of their research in terms of weeks and class hours. In some studies, only information on the amount of weeks was given; in others, only information on the class hours was provided. In some studies, both week and class hour information was included, while one study lacked any information about the duration of its research. After the analysis, it was determined that 22 studies were conducted over five weeks, 21 studies lasted eight weeks, 16 studies lasted four weeks, 13 studies lasted nine weeks, 11 studies lasted six weeks, and 9 studies lasted three weeks. The study durations related to other elements can be examined from the decreasing Pareto diagram. Based on Figure 11, it was concluded that in 80% of the studies conducted to determine the effects of problem-based learning, the research was carried out over periods of 5, 8, 4, 9, 6, and 3 weeks.

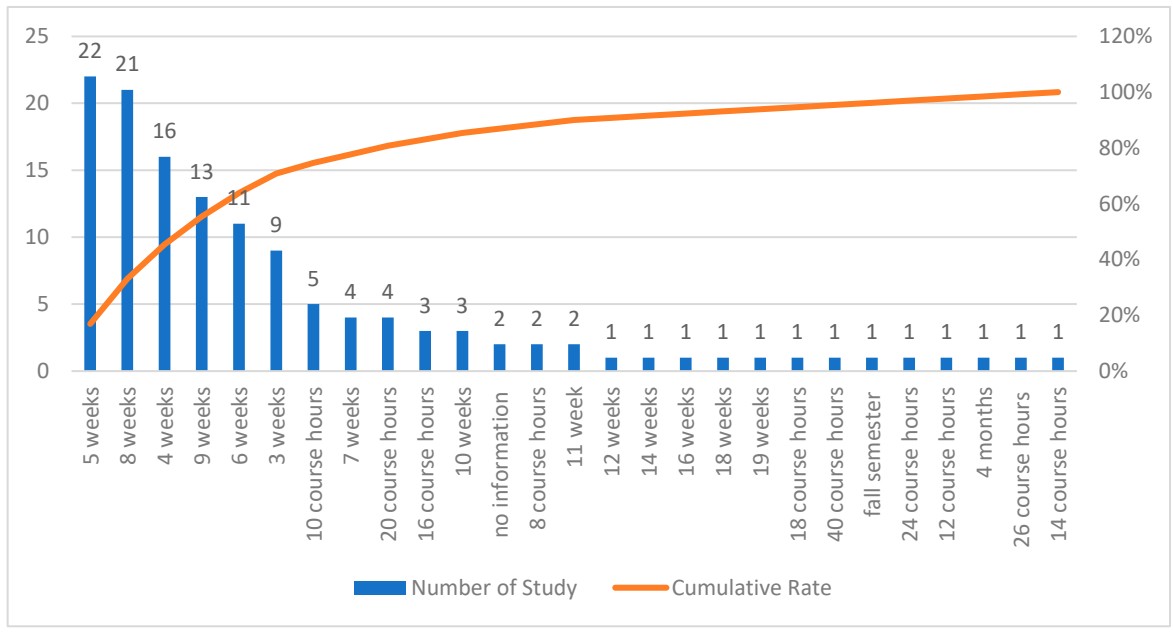

**Figure 11.** Pareto diagram of findings regarding the duration of the research.

*5.9. Findings Regarding the Results of Reviewed Studies*

When Figure 12 is examined, it can be determined that the PBL method had a positive effect on academic achievement (76), attitude (35), science process skills (17), students' opinions (18), conceptual understanding (14), problem-solving skills (12), motivation (12), creative thinking skills (9), knowledge retention (8), overcoming misconceptions (7), critical thinking skills (4), self-efficacy beliefs (3), and developing environmental awareness (3). In all of the conducted studies, PBL had a neutral effect, while in 47 studies conducted with these variables, including attitude (18), academic achievement (10), problem-solving skills (5), science process skills (4), motivation (3), knowledge retention (2), creative thinking skills (1), self-efficacy beliefs (1), inquiry learning skills (1), metacognition (1) and reflective thinking (1), it was found that it had no effect. In none of the examined studies was a negative effect of problem-based learning method detected.

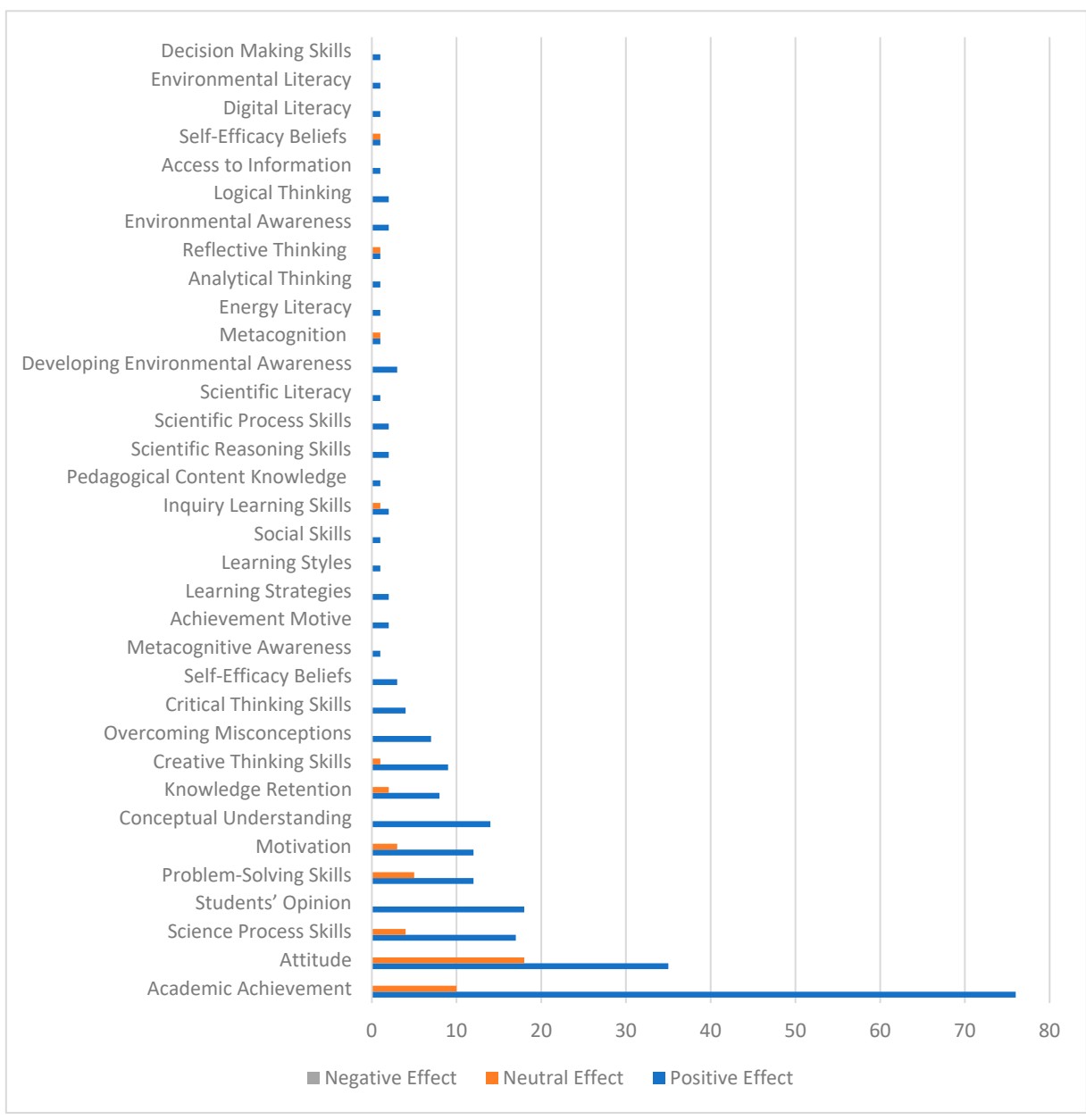

**Figure 12.** The findings of the reviewed studies' results.

*5.10. Findings Regarding the Science Topics in the Examined Studies*

When Table 2 is examined, it can be determined that research was conducted on topics related to physics, chemistry, biology, the environment, and astronomy. The effectiveness of the PBL method was mainly studied in physics, while there was a limited number of studies in the fields of the environment and astronomy. When all of the topics are considered, the most researched topics appear to be "Matter and Heat" (15), "Force and Motion" (14), "Electricity and Magnetism" (12), "Work-Energy" (9), "Acid-Base" (9), and "Systems in Our Body" (8).

**Table 2.** Distribution of science topics in the examined studies.

| Discipline | Subjects | f |
|---|---|---|
| Physics | Matter and Heat | 15 |
| | Force and Motion | 14 |
| | Electricity and Magnetism | 12 |
| | Work–Energy | 9 |
| | Light and Sound | 4 |
| | Pressure | 3 |
| | States of Matter and Heat Fuels | 2 |
| | Light and Optics | 2 |
| Chemistry | Acid–Base | 9 |
| | Mixtures | 6 |
| | Particulate Structure of Matter | 6 |
| | Electrolysis and Batteries | 6 |
| | Chemical Properties of Solutions | 3 |
| | Physical and Chemical Changes | 2 |
| | Atomic and Periodic System | 1 |
| Biology | Systems in Our Body | 8 |
| | Heredity/Genetics | 4 |
| | Reproduction, Growth, and Development in Plants and Animals | 4 |
| | Matter Cycles | 1 |
| | Basic Structure of the Cell | 1 |
| | Natural Systems | 1 |
| | Viruses and Bacteria | 1 |
| | Bacteria and Archaea | 1 |
| Environment | Humans and the Environment | 4 |
| | Environmental Science | 2 |
| | Why do Ecosystems Change? | 1 |
| | Environmental Problems | 1 |
| | Energy Transformations | 1 |
| | Domestic Waste and Recycling | 1 |
| Astronomy | Astronomy and Renewable Energy | 1 |
| | Solar System and Eclipses | 1 |

## 6. Discussion and Conclusions

In this study, research focusing on PBL in science education in Türkiye was examined using a descriptive content analysis method. The studies were scrutinized in detail, and by conducting a critical review of the data from these studies, general trends were identified. The studies conducted in Türkiye include meta-analyses, content analyses, descriptive content analyses, and systematic literature reviews related to PBL [6,22,24,25,33,34]. In some of these studies, "achievement" [33,35], "achievement and attitude" [36], and "achievement, attitude, motivation, and group dynamics" [34] were examined, while in others, the general outlines of PBL were explored [22,24,25]. In our study, we also examined the studies related to PBL in science education in a broad context.

As per the objective of this study, a total of 133 works related to PBL in science in Türkiye, comprising 96 graduate theses and 37 articles published between 2000 and 2023,

have been examined. It was noted that there were three separate content analysis studies conducted on previous PBL research in science in Türkiye [24,25,35]. Additionally, six different content analysis studies were identified that examined PBL without limiting the scope to a specific field of science [22,33,34,36,37]. In their respective works, Alper et al. [22] examined 48 studies, Batdi [33] examined 26, Dagyar [37] examined 118, Ayaz [35] examined 24, Temel et al. [24] examined 58, Tosun and Yasar [25] examined 40, Mutlu and Aydogmuş [6] examined 40, Ozgul [36] examined 107, and Bati [34] examined 20 different studies. Therefore, it can be argued that our study is more comprehensive than previous similar studies.

In our study, when the distribution of the types of studies is examined, it is determined that 49.6% of the studies are master's theses, 27.8% are articles, and 22.5% are doctoral theses. When the studies are examined by publication year, it is observed that no studies were conducted between 2000 and 2003. The first studies related to PBL in science in Türkiye seem to have started in the year 2003. Additionally, it is noted that the highest number of studies were conducted in 2012 and 2017. It is found that 75,18% of the studies were conducted between 2010 and 2021. In 2022 and 2023, three studies were conducted each year, leading to the conclusion that the number of studies related to PBL in science has significantly decreased in recent years. Similarly, Mutlu and Aydogmuş [6] reported a significant decrease in studies related to PBL in Türkiye in recent years based on their examination of 40 theses conducted between 2014 and 2018.

Our results show that, in the conducted studies, 69 different dependent variables were examined. In studies related to PBL in science, the variable most frequently investigated was the impact of the method on achievement (64.61%). Additionally, attitude, scientific process skills, and problem-solving skills were the next most commonly examined variables, in that order.

Dagyar [37] conducted a meta-analysis of 118 studies in his doctoral thesis and examined the impact of the PBL approach on academic achievement. Similarly, Ozgul [36] conducted a meta-analysis study examining the effects of PBL on students' academic achievement and attitudes in 107 studies. Ozgul [36] found that in the studies he reviewed, the PBL approach had a positive impact on academic achievement in 73 cases and a negative impact in 4 cases. Furthermore, Batdi [33] in his study determined that online PBL had a positive effect on students' achievements, attitudes, and motivation. Based on this, it can be concluded that the use of PBL in classes has a positive impact on academic achievement. Additionally, Temel et al. [24] reported in their study that academic risk-taking, metacognitive awareness, logical thinking ability, learning strategies, the permanence of learned knowledge, problem-solving skills, creativity, motivation, and scientific process skills were also examined in research studies.

In this study, we also found that the PBL method had a positive impact on 34 different skills, and it had no impact on 11 different skills. Additionally, there was no evidence of the method having a negative impact on any skill. More than half of the studies (53.38%) identified a positive impact of the method on achievement. Furthermore, in 35 studies, the applied method had a positive effect on attitude, while in 18 studies, it had no effect on attitude. In 17 studies, the applied method had a positive impact on scientific process skills; in 18 studies, it had a positive impact on student opinions; in 14 studies, it had a positive impact on conceptual understanding; and in 12 studies, it had a positive impact on problem-solving skills.

Our results show that the most preferred research design was the quasi-experimental design in studies related to PBL in science. This was followed, in order, by the full experimental design, mixed-method design, weak experimental design, case study approach, action research approach, and weak experimental design. The analysis revealed that quantitative methods were more commonly used, with qualitative methods being less preferred. Additionally, it was observed that the earliest studies related to PBL in science in Türkiye were full experimental studies. The highest number of full experimental studies appeared in 2010, while the highest number of quasi-experimental studies occurred in 2017 and 2020.

The first qualitative action research study in science education was conducted in 2008. It was noted that no mixed-method research was conducted until 2010, but there have been 13 mixed-method research studies conducted since 2017. Consequently, it can be inferred that 76.47% of the mixed-method research studies were conducted in recent years. In their study, Mutlu and Aydogmuş [6] reported that experimental designs were predominantly preferred in the conducted studies.

When the studies related to PBL in science were examined according to the level of education that was studied, the findings showed that more than half of the studies (56.39%) were conducted at the middle school level. Following the middle school level, studies conducted at the undergraduate level had the next highest number (27.06%). The analysis revealed that there were sixteen studies conducted at the high school level, three studies conducted at the primary school level, and one study conducted at the preschool level. Furthermore, it was observed that the majority of the studies at the middle school level were conducted with seventh-grade students, followed by sixth-grade and fifth-grade students. When the studies conducted at the undergraduate level were examined, it was found that the sample predominantly consisted of third-year students.

Another noteworthy result obtained from this descriptive content analysis is the limited inclusion of final-year students in the samples at various school levels. This could be attributed to examination attempts such as the High School Transition System exam in the 8th grade, the Higher Education Institutions Examination in the 12th grade, and the Public Personnel Entrance Exam held in the senior year of undergraduate studies for students wishing to start a professional career. Mutlu and Aydogmuş [6] reported that in their examination of 40 theses related to PBL conducted between 2014 and 2018 in Türkiye, middle school students were predominantly selected as samples, and they did not come across any studies that included primary school students. Similarly, Tosun and Yasar [25] noted in their study that primary school students were selected as samples in research studies to a low extent, and there were no samples selected from preschool education.

As a result of this study, it was observed that 68.42% (26) of the studies conducted at the undergraduate level were related to prospective science teachers. Furthermore, the analysis revealed that in the research studies, 31 different topics from physics, chemistry, biology, environmental science, and astronomy were addressed, with 8 of them being related to physics, 7 to chemistry, 8 to biology, 6 to environmental science, and 2 to astronomy. When considering all of the topics covered in the studies, it was determined that the most frequently researched topics were, in order: "Matter and Heat", "Force and Motion", "Electricity and Magnetism", "Work-Energy", and "Acid-Base". In their study, Temel et al. [24] reported that 20 studies were conducted within the scope of middle school science courses, and 16 studies were conducted within the scope of undergraduate chemistry courses. Additionally, consistent with the findings obtained in the present study, Tosun and Yasar [25] found that the most preferred topic in their study was "force and motion", whereas in our study, the most preferred topics were "matter and heat" and "force and motion".

After the analysis, it was determined that 22 studies were conducted over five weeks, 21 studies lasted eight weeks, 16 studies lasted four weeks, 13 studies lasted nine weeks, 11 studies lasted six weeks, and 9 studies lasted three weeks.

This descriptive content analysis study underscores the growing interest in problem-based learning (PBL) within the context of science education in Türkiye. Despite challenges associated with resistance to change and faculty development, PBL in Türkiye has shown promise in improving students' critical thinking, problem-solving skills, and subject matter comprehension. As Türkiye continues to explore PBL, further research should focus on addressing these challenges and assessing the long-term effects of PBL on science education in the country.

While the articles reviewed here reflected on the challenges of PBL, they also pointed to future directions for PBL in science education in Türkiye. There was a consensus on the importance of faculty development programs to prepare educators for the transition to PBL.

Additionally, this research emphasized the need for further investigation into the long-term impact of PBL on students' learning outcomes and their preparedness for the workforce.

**Author Contributions:** Conceptualization, B.A.; methodology, B.A. and İ.B.; validation, B.A. and İ.B.; formal analysis, B.A. and İ.B.; investigation, B.A. and İ.B.; resources, İ.B.; data curation, İ.B.; writing—original draft preparation, B.A. and İ.B.; writing—review and editing, B.A. and İ.B.; supervision, B.A.; project administration, B.A. All authors have read and agreed to the published version of the manuscript.

**Funding:** This research received no external funding.

**Institutional Review Board Statement:** Not applicable.

**Informed Consent Statement:** Not applicable.

**Data Availability Statement:** No new data were created.

**Conflicts of Interest:** The authors declare no conflicts of interest.

## Appendix A

Bibliographies of the studies in alphabetical order

Acikyildiz, M. (2004). *Investigation of effectiveness of problem-based learning at physical chemistry laboratory experiments* (Master's Thesis). Atatürk University, Erzurum.

Akbulut, H.H. (2010). *Implementation and evaluation of problem based learning on buoyant force and floating concepts* (Master thesis). Karadeniz Technical University, Trabzon.

Akcay, B. (2009). Problem-based learning in science education. *Journal of Turkish Science Education*, *6*(1), 28–38.

Akin, S. (2008). *Teaching environmental problems caused by stubble fires, ozon layer depletion and vehicles through problem-based learning* (Master thesis). Atatürk University, Erzurum.

Aksar B.G. (2022). *The effect of the problem based learning method applied in distance education on the academic achievement, conceptual understanding and attitude of the environment on the human and environment unit* (Master thesis). Marmara University, Istanbul.

Akti Aslan, S. (2019). *The effect of virtual learning environments designed according to problem-based learning approach to students' success, problem-solving skills, and motivations* (Doctoral dissertation). Inonu University, Malatya.

Alici, M. (2018). *Effect of STEM instruction on attitude career perception and career interest in a problem based learning environment and student opinions* (Master thesis). Kırıkkale University, Kırıkkale.

Alper, Y.A. (2003). *The Effect of cognitive flexibility on students achievement and attitudes in web mediated problem based learning* (Doctoral dissertation). Ankara University, Ankara.

Altuncekic, A. (2010). *The effect of web supported problem based learning medium upon cognitive and efffective learning products:* Gazi University Kastamonu Education Example (Doctoral dissertation). Gazi University, Ankara.

Arici, F. (2021). *Investigation of the effectiveness of problem based learning method of enhanced with augmented reality in the teaching of cell and divisions subject* (Doctoral dissertation). Atatürk University, Erzurum.

Arslan Turan B. (2014). *Effects of problem-based learning on achievement, self-regulated learning skills and academic self-concept* (Doctoral dissertation). Hacettepe University, Ankara.

Arslan, A. (2009). *The effect of learning style based on problem on the success of student in studying human and environment unit* (Master thesis). Sakarya University, Sakarya.

Ayaz, G. (2007). *The effect of problem-based learning on the elementary school students' achievement in genetics* (Master thesis). Middle East Technical Unıversity, Ankara.

Aydogdu, C. (2012). The effect of problem based learning strategy in electrolysis and battery subject teaching. *Hacettepe University Journal of Education*, *42*(42), 48–59.

Aysu, G. (2019). *An examination of the effects of problem-based learning STEM applications on students' academic achievements and permanence of learning* (Master thesis). Niğde Ömer Halisdemir University, Niğde.

Ayyildiz, Y. (2012). *Development, application and evaluation of an active learning material based on constructivist approach in the subject of chemical reactions and energy in the high school chemistry* (Doctoral dissertation). Dokuz Eylül University, Izmir.

Balim, A.G., Celiker, H.D., Turkoguz, S., Evrekli, E., & Ekici, D. İ. (2015). The effect of concept cartoons-assisted problem-based learning method on conceptual understanding levels and problem solving skill perceptions of students. *Journal of Turkish Science Education*, *12*(4), 53–76.

Balim, S. (2016). *The effects of using problem based learning method in science courses on talented students' levels of academic achievements, inquiry learning skill perceptions and attitudes towards science* (Master thesis). Dokuz Eylul University, Izmir.

Bayrak, B.K., & Bayram, H. (2012). The effect of problem-based learning in a web environment on the academic achievement of students with different learning styles. *Mustafa Kemal University Journal of Social Sciences Institute*, *9*(18), 479–497.

Bayram, A. (2010). *The effect of problem based learning on overcoming 5th grade students' misconceptions about "heat and temperature"* (Master thesis). Selcuk University, Konya

Bayram, F. (2023). *The effect of problem based learning approach to teaching the unit of the solar system and eclipses on academic achievement and problem solving skills* (Master thesis). Kocaeli University, Kocaeli.

Benli, E. (2010). *The research of the effects of problem based learning to the permanence of information, the academic success of science teacher candidates and their attitudes toward science* (Master thesis). Gazi University, Ankara.

Boncukcu, G. (2020). *The effect of problem based learning model on environmental literacy, problem solving and self-regulatory skills of 8th grade students in sustainable development* (Master thesis). Mersin University, Mersin.

Buyukdokumaci, H. (2012). *Effects of problem based learning on learning products in science and technology lesson for elementary 8th grade* (Master thesis). Pamukkale University, Denizli.

Can, I. (2021). *Determining of the Effects Problem-Based Science Learning Activities on Gifted Students* (Master thesis). Uşak University, Uşak.

Cayan, Y., & Karsli, F. (2015). The effects of the problem based teaching learning approach to overcome students' misconceptions on physical and chemical change. *Kastamonu Education Journal*, *23*(4), 1437–1452.

Celik, E. (2010). *The effect of problem based learning approach in science education on students? academic achievement, attitude, academic risk taking level and retention of knowledge* (Master thesis). Gazi University, Ankara.

Cakici, Y., Soyleyici, H., & Oguzhan Dincer, E. (2020). Investigation of the effect of problem based learning on the scientific process skills, attitudes and achievements of middle school students': unit light. *Trakya University Journal of Social Science*, *22*(1), 419–437.

Celik, E., Eroglu, B., & Selvi, M. (2012). The effect of problem based learning approach in science education on students' academic achievement, and attitudes toward science and technology course. *Kastamonu Education Journal*, *20*(1), 187–202.

Dadli, G. (2017). *The effect of learning activities based on authentic problems in the human and environmental relations unit on the reflective thinking skills, academic success, environmental attitude and awareness of 7th grade students* (Master thesis). Kahramanmaraş Sütçü İmam University, Kahramanmaraş.

Demirel, M., & Turan, B. (2010). The effects of problem based learning on achievement, attitude, metacognitive awareness and motivation. *Hacettepe University Journal of Education*, *38*, 55–66.

Demirel, O.E. (2014). *Effects of problem based learning and argumentation based learning on the students' chemistry achievement, their science process skills and science reasoning attitudes* (Master thesis). Mustafa Kemal University, Hatay.

Divarci, Ö. F. (2016). *The effects of the multimedia supported problem based learning on 8th grade students in relation to academic success, attitude and permanence: The subject of pressure* (Master thesis). Amasya University, Amasya.

Divarci, Ö. F., & Saltan, F. (2017). The effects of the multimedia supported problem based learning on academic success and attitude in science education. *Journal of Kırşehir Education Faculty*, *18*(3), 1–23.

Dursun, C. (2015). *The effect of problem-based learnıng method on students' attitude towards the environment and their environmental awareness* (7th Grade "Human and Environment" Unit Example) (Master thesis). Pamukkale University, Denizli.

Eceyurt Turk, G. (2017). *The effect of argumentation-supported problem based learning applications on the acid/bases and gases success of pre-service science teachers* (Doctoral dissertation). Gazi University, Ankara.

Elbistanli, A. (2012). *Investigation of the effect of problem based learning approach on the achievement, attitude and scientific process skills of 11. grade students through chemical equilibrium subject* (Master thesis). Mustafa Kemal University, Hatay.

Erden, S., & Yalcin, V. (2021). Investigation of the effect of preschool stem activities prepared according to the problem based learning approach on children's problem-solving skills. *Trakya Journal of Education*, *11*(3), 1239–1250.

Erdogan, E.M. (2022). *Investigation of problem-based learning method, enriched with the 5E learning model global warming and its effect on attitudes on climate change* (Master thesis). Kırıkkale University, Kırıkkale.

Eren, C.D. (2011). *The effect of problem based learning (PBL) on critical thinking disposition, concept learning and scientific creative thinking skill in science education* (Doctoral dissertation). Marmara University, Istanbul.

Eyceyurt Turk, G., & Kilic, Z. (2020). The effect of argumentation-supported problem based learning on the achievements of science teacher candidates regarding the subjects of gases and acids-bases. *Bartın University Journal of Faculty of Education*, *9*(2), 440–463.

Fettahlioglu, P. (2012). *The usage of argumentation-based and problem-based learning approaches intended for developing the environmental literacy of pre-service science teachers* (Doctoral dissertation). Gazi University, Ankara.

Fidan, M. (2018). *The impact of problem-based science teaching assisted with augmented reality applications on academic achievement, retention, attitude and belief of self-efficacy* (Doctoral dissertation). Bolu Abant İzzet Baysal University, Bolu.

Germi, N.T. (2020). *The effect of problem based learning 5th grade students' change of substance unit on achievements, creative thinking skills, comprehension levels of concepts and their motivations* (Doctoral dissertation). Ondokuz Mayıs University, Samsun.

Gocuk, A. (2015). *To improve the energy literacy for the 5th grades by using the approach of problem based learning* (Master thesis). Marmara University, Istanbul.

Gogus, R. (2013). *Problem based learning teaching science and its effect on students' academic achievement and attitudes* (Master thesis). Kırıkkale University, Kırıkkale.

Gokmen, S. İ. (2008). *Effects of problem based learning on students' environmental attitude through local vs. non local environmental problems* (Master thesis). Middle East Technical University, Ankara.

Gunter, T., & Alpat, S.K. (2018). Students' opinions about problem-based learning and scenario applied in teaching 'electrochemistry'. *Karaelmas Fen ve Mühendislik Dergisi*, *8*(1), 346–358.

Guzel, Z. (2018). *The effects of problem based approach practiced through self and peer assessment on students' achievements and attitudes in science teaching* (Master thesis). Necmettin Erbakan University, Konya.

Gül, Ş., & Konu, M. (2018). The effect of context- and problem-based learning activities on the students' achievements. *Education for Life*, *32*(1), 45–68.

Hiğde, E. & Aktamış, H. (2023). Students' views on problem-based blended learning during the Covid-19 pandemic process. *Trakya Journal of Education*, *13*(1), 260–279

Hun, F. (2017). *The effect of academic achievement and attitudes on the 7th grade students of problem based learning method and improved 5E learning model* (Master thesis). Giresun University, Giresun.

Hun, F., & Degirmencay, S.A. (2020). Effect of 5E teaching model supported by problem based learning on the achievement and attitude. *OPUS International Journal of Society Researches*, *16*(29), 1689–1717.

Ince Aka, E. (2012). *The effect of problem-based learning method used for teaching acids and bases on different variables and students? views on the method* (Doctoral dissertation). Gazi University, Ankara.

Inel, D. (2009). *The effects of the using of problem based learning method in science and technology course on students? the levels of constructing concepts, academic achievements and enquiry learning skill perceptions* (Master thesis). Dokuz Eylul University, Izmir.

Inel, D. (2012). *The effects of concept cartoons-assisted problem based learning on students? problem solving skills perceptions, motivation toward science learning and levels of conceptual understanding* (Doctoral dissertation). Dokuz Eylül University, Izmir.

İnaltekin, T., & Şahin, F. (2019). The effects of problem based learning on development of preservice science teachers' pedagogical content knowledge. *Ege Journal of Education*, *20*(1), 78–112.

Ince Aka, E. & Sarikaya, M. (2015). The effect of problem-based learning method on attitudes of preservice science teachers towards chemistry lesson. *GUJGEF, 34*(3), 451–467.

Ince Aka, E., & Sarikaya, M. (2017). Developing attitude scale toward the problem based learning method. *Journal of Kırşehir Education Faculty*, *18*(2), 75–95.

Kacar, S. (2012). The *effects of problem based learning method integrated visual arts on students? academic achievements, scientific creativities and attitudes towards science teaching with art activities* (Master thesis). Dokuz Eylul university, Izmir.

Kacar, S., Ormanci, U., Ozcan, E., & Balim, A.G. (2020). Concept cartoon samples integrated into problem based learning in a science course. *Journal of Inquiry Based Activities*, *10*(2), 127–145.

Kan, S. (2013). *Evaluating project-based and problem-based instructional applications in physics instruction* (Doctoral dissertation). Karadeniz Technical University, Trabzon.

Kanar, A. (2017). *Examination of the effects of the using of problem based learning method in the teaching science and laboratory practice course on pre-service science teachers* (Master thesis). Uşak University, Uşak.

Kanar, A., & Inel Ekici, D. (2020). Problem-based science learning implementations integrated with designing activity with pre-service teachers. *Turkish Studies-Educational Sciences*, *15*(1), 549–570.

Kanli, E. (2008). *The effect of problem based learning in science & technology instruction on gifted and normal students' achievement, creative thinking and motivation levels* (Master thesis). Istanbul University, Istanbul.

Kanli, E., & Emir, S. (2013). The effect of problem based learning on gifted and normal students' achievement and creativity levels. *Necatibey Faculty of Education Electronic Journal of Science & Mathematics Education*, *7*(2), 18–45.

Karagoz, M.P. (2008). *The effect of teaching the unit of "power and motion" in primary school science course using the problem based learning approach on students science process skills, success and attitude* (Master thesis). Mugla University, Mugla.

Kartal Tasoglu, A. (2009). *The effect of problem based learning on students' achievements, scientific process skills and attitudes towards problem solving in physics education* (Master thesis). Dokuz Eylul University, Izmir.

Kartal Tasoglu, A. (2015). *Investigating the effects of the problem based learning approach on understanding the topics of magnetism* (Doctoral dissertation). Dokuz Eylül University, Izmir.

Keles, M. (2015). *The effect of problem-based learning method on student' recall level and success in the processing of science and technology course* (Master thesis). Necmettin Erbakan University, Konya.

Kilic, I., & Moralar, A. (2015). The effect of problem-based learning approach on academic success and motivation in science education. *Pegem Journal of Education and Instruction*, *5*(5), 625–636.

Kizilcik, h. S. (2012). *A case study on development of heat and temperature concepts in process of problem-based learning* (Doctoral dissertation). Gazi University, Ankara.

Kizilkaya, A. (2017). *The effect of problem based learning and jigsaw I technic in science teaching on students' academic achievement and knowledge permanence in steps of bloom taxonomy cognitive domain* (Doctoral dissertation). Atatürk University, Erzurum.

Kocakoglu, M. (2008). *The effect of problem based learning and motivational styles on students' academic success and attitudes towards biology course* (Doctoral dissertation). Gazi University, Ankara.

Konu, M. (2017). *The effect of context and problem based instruction on the students 'achievements, attitudes, motivations and problem solving skills in biology course* (Doctoral dissertation). Atatürk University, Erzurum.

Korucu, N.E. (2007). *Comparing with problem and cooperative based learning method applied in primary schools on the success of the students* (Master thesis). Selçuk University, Konya.

Kulekci, E. (1019). *Effects of concept cartoon assisted problem based Science, Technology, Engineering and Mathematics (STEM) activities on 5th grade science teaching* (Master thesis). Manisa Celal Bayar University, Manisa.

Kumas, A. (2008). *An assessment and implementation of problem based learning in cooperative learning groups in the unit of motion on the earth* (Master thesis). Karadeniz Technical University, Trabzon.

Kumbasar, T. (2019). *The effect of problem-based teaching intervention about acids and bases over the learning of students with different learning styles and multiple intelligence* (Master thesis). Marmara University, Istanbul.

Kurt, U. (2020). *Comparison of the effects of different active learning methods on the teaching of 'Cell and divisions' and 'Force and energy' units* (Doctoral dissertation). Atatürk University, Erzurum.

Kusdemir, M. (2010). *An analysis of the effect of problem based learning model on the students succes, attitude and motivations* (Master thesis). Mustafa Kemal University, Hatay.

Kusdemir, M., Ay, Y., & Tuysuz, C. (2013). An analysis of the effect of problem based learning model on the 10 th grade students' achievement, attitude and motivation in the unit of "mixtures". *Necatibey Faculty of Education Electronic Journal of Science & Mathematics Education*, *7*(2), 195–224.

Kuvac, M. (2014). *The effects of problem based learning on preservice Science Teachers' environmental consciousness and metacognitive awareness* (Master thesis). Istanbul University, Istanbul.

Kucuk Avci, S. (2017). *The effect of problem based learning in 3 dimensional virtual learning environments on conceptual understanding and learning performance based on problem solving* (Doctoral dissertation). Sakarya University, Sakarya.

Moralar, A. (2012). *The effect of problem-based learning approach on academic success, attitude and motivation in science education* (Master thesis). Trakya University, Edirne.

Nerse, B.N. (2021). *Study of the effect of the problem-based learning approach enriched with web 2.0 tools on the academic achievement, metacognitive awareness, self-directed learning with technology and digital literacy of students during online education process* (Master thesis). Kocaeli University, Kocaeli.

Olca, M. (2015). *The effects of problem based learning method on students' analytical thinking skills, conceptual understandings and attitudes toward science* (Master thesis). Dokuz Eylul University, Izmir.

Ozcan, E. (2013). *Effects of problem based learning on prospective science teachers' problem solving skills, academic achievements and attitudes* (Master thesis). Dokuz Eylul University, Izmir.

Ozcelik, C. (2021). *The impact of problem-based STEM applications on students' attitudes towards STEM, self-regulation skills and metacognitive abilities* (Doctoral dissertation). Bartın University, Bartın.

Ozdeniz, Y. (2021). *The effect of application of science module designed according to integrated curriculum model in blended learning environment on scientific reasoning and scientific process skills of gifted students* (Master thesis). Aydın Adnan Menderes University, Aydın.

Ozeken, O.F. (2011). *An investigation of effectiveness of problem based learning in teaching acid-base subject* (Doctoral dissertation). Atatürk University, Erzurum.

Ozkan, E. (2021). *Evaluation of the effectiveness of the problem based teaching model on the students' achievements and attitudes in high school chemistry courses* (Master thesis). Atatürk University, Erzurum.

Ozturk, Z.D. (2019). *The effect of problem based learning method on students' academic achievements and scientific process skills in a science course* (Master thesis). Pamukkale University, Denizli.

Ozturk, Z.D., & Ozel, M. (2021). The effect of problem based learning on students' scientific process skills. *The Journal of Buca Faculty of Education*, *51*, 1–31.

Saglam, O. (2022). *The effect of the argumentation-supported problem-based learning method on the teaching of the topic of 8th grade "simple machines"* (Master thesis). Sivas Cumhuriyet University, Sivas.

Sagir Ulucinar, Ş., Celik Yalcin, A., & Armagan Oner, F. (2009). The effect of problem based learning strategy in metalic activity subject teaching. *Hacettepe University Journal of Education*, *36*, 283–293.

Sahbaz, O. (2010). *The effects of different methods on students' science process skills, problem solving skills, academic achievements and retentions in primary school fifth grade science and technology lessons* (Doctoral dissertation). Dokuz Eylül University, Izmir.

Sahin, A. (2011). *To Analyze the effect of problem based learning (PBL) approach on academic success of students in teaching basic electrical circuits in general physics laboratory course* (Master thesis). Atatürk University, Erzurum.

Salgam, E. (2009). *The effect of problem based learning method on students' academic achievement and their attitude on physics education* (Master thesis). Dokuz Eylul University, Izmir.

Sarikaya, S.(2006). *Interactive teaching methods in environmental education* (Master's Thesis). Celal Bayar University, Manisa.

Sencan, D. (2013). *The effects of real-world problems on the 7th grade students' scientific process abilities, academic achievement and scientific literacy: Force and motion* (Master thesis). Marmara University, Istanbul.

Serin, G. (2009). *The effect of problem based learning instruction on 7th grade students' science achievement, attitude toward science and scientific process skills* (Doctoral dissertation). Middle East Technical University, Ankara.

Sifoglu, N. (2007). *The effects of constructivism and problem-based learning on students' success in the teaching the topic heritage" at the 8th grade* (Master thesis). Gazi University, Ankara.

Soyleyici, H. (2018). *Investigating the effect of problem-based learning on the scientific process skills, scientific attitudes, achievements and conceptual knowledge of middle school students': Unit light* (Master thesis). Trakya University, Edirne.

Senocak, E., & Taskeseligil, Y. (2005). Problem-Based Learning and its Applicability in Science Education. *Kastamonu Education Journal*, *13*(2), 359-366.

Tasoglu, A.K., & Bakac, M. (2023). The effect of problem based learning approach on students' academic achievement and critical thinking skills. *The Journal of National Education*, *52*(238), 855–884.

Tatar, E., Oktay, M., & Tuysuz, C. (2009). Advantages and disadvantages of problem based learning in chemistry education: A case study. *Journal of Education Faculty*, *11*(1), 95–110.

Tavukcu, K. (2006). *The effects on the learning outcomes of problem based learning in science instruction* (Master thesis). Zonguldak Karaelmas, Zonguldak.

Tekin, A.D. (2019). *Academic success of 7th grade students of problem based learning approach impact on process skills and motivations* (Master thesis). Marmara University, Istanbul.

Tosun, C. (2010). *The effect of problem based learning method on understanding of the solutions and its? physical properties* (Doctoral dissertation). Atatürk University, Erzurum.

Tosun, C., & Taskesenligil, Y. (2012). The effect of problem based learning on student motivation towards chemistry classes and on learning strategies. *Journal of Turkish Science Education*, *9*(1), 36–50.

Tosun, C., Senocak, E., & Ozeken, O.F. (2013). The effect of problem-based learning on undergraduate students' learning about solutions and their physical properties and scientific processing skills. *Chemistry Education Research and Practice*, *14*(1), 36–50.

Tunc, T. (2015). *The effect of problem based learning on students' academic achievements in the subject of Electrochemistry in Analytical Chemistry course* (Doctoral dissertation). Dokuz Eylül University, Izmir.

Tuysuz, C., & Demirel, O.E. (2020). *Examining the Effects of Problem Based and Argumentation Based Learning Methods on the "Mixtures" Unit. MSKU Journal of Education*, *7*(1), 43–61.

Ulukok, S. (2012). *The effect of computer assisted problem based learning method on the higher level thinking skills of prospective teachers* (Master thesis). Kırıkkale University, Kırıkkale.

Unal, S.Y. (2019). *The effect of problem based learning approach on teaching the subject 'the reproduction, growth and development of plant and animal'* (Master thesis). Uşak University, Uşak.

Vekli, G.S., & Cimer, A. (2017). Effect of problem-based computer-aided material (pbcam) on development of students' perceived problem solving skills. *Bayburt Faculty of Education Journal*, *12*(24), 809–830.

Yalcinyigit, C. (2016). *Research about critical thinking in problem based learning on biology* (Doctoral dissertation). Gazi University, Ankara.

Yaman, S. (2003). *The effects on the learning outputs of problem based learning in science education* (Doctoral dissertation). Gazi University, Ankara.

Yaman, S., & Yalcin, N. (2005). Effectiveness of problem-based learning approach on development of problem solving and self-efficacy beliefs levels in science education. *Hacettepe University Journal of Education*, *29*(2005), 229–236.

Yaman, S., & Yalcin, N. (2005). Investigating of the effects on academic achievement and creativity thinking skills of prospective teachers' of problem based learning. *Bolu Abant Izzet Baysal University Journal of Faculty of Education*, *5*(2), 108–121.

Yildirim, C. (2017). *The effects of argumentation supported problem based learning on students' inquiry learning skills and problem solving skills and levels of conceptual understanding* (Doctoral dissertation). Pamukkale University, Denizli.

Yildirim, C., & Can, B. (2018). The effects of argumentation supported problem based learning on students' inquiry learning skill perceptions. *PAU Journal of Education*, *44*(44), 251–277.

Yildirim, H. (2011). *The effect of problem based and project based learning styles on primary school students? successes and attitude* (Master thesis). Selcuk University, Konya.

Yildiz, E., Simsek, U., & Yuksel, F. (2017). The effect of jigsaw-integrated problem based learning method on students' motivation towards science learning, social skills and attitude towards school. *Kastamonu Education Journal*, *25*(5), 1957–1978.

Yildiz, N. (2010). *The effect of experiment applications on the success, attitude and scientific process abilities of the students in the solution of the learning scenarıos based on problems in science education* (Master thesis). Marmara University, Istanbul.

Yildiz, S. (2019). *The investigation of the effect of problem based teaching approach on students' problem solving skills and academic achievements in science course* (Master thesis). Sakarya University, Sakarya.

Yildiz, T. (2017). *Investigation of the effectiveness of problem based teaching method in teaching the particulate structure of matter unit in primary science course* (Master thesis). Ağrı İbrahim Çeçen University, Ağrı.

Yilmaz, E. (2015). *The effect of teaching methods supported with online advance organizer concept teaching material to achievement, attitude and retention in force-motion unit* (Master thesis). Mehmet Akif Ersoy University, Burdur.

Yilmaz, T. (2016). *Problem based learning method's effect on teaching science education courses for secondary school 5. grade students' academic achievements and attitudes towards science education classes: Light and sound* (Master thesis). Bozok University, Yozgat.

Yilmazel, A. (2020). *The effect of the problem-based learning method in eliminating the misconception of heat and temperature of secondary school students* (Master thesis). Mersin University, Mersin.

Yurd, M. (2007). *The effect of know-want-sample-learn strategy, which is developed by using problem based learning and know-want-learn strategy, towards the 5th grade students' attitudes in science and technology lesson and towards to remove their misconceptions* (Master thesis). Mustafa Kemal University, Hatay.

Yurd, M., & Olgun, O.S. (2008). Effect of problem based learning and know-want-learn strategy to remove misconceptions. *Hacettepe University Journal of Education*, *35*, 386–396.

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
