# Peer review of "Problem-Based Learning in Türkiye: A Systematic Literature Review of Research in Science Education"

_education, doi:10.3390/educsci14030330_

Round 1

Reviewer 1 Report

Comments and Suggestions for Authors

The study was presented clearly and was easy to follow. It is certainly publishable once the following corrections are addressed.

What was the inter-rater reliability coefficient? Please provide the value.

Line 452 – problem-solving skills

Line 522: what do you mean with permanence in this context?

Line 576- acids and bases are presented twice

Line 580: Yasar [21] noted in their research that the most preferred topic in the studies was 'force and motion.' This is different from your study as yours are Matter and heat – so please comment on this

A more serious comment is that you did pareto diagrams. Therefore, in your recommendations you can specify areas to focus on according to the 80 – 20 principle. Since you possess the data, its essential to provide recommendations and commentary based on your findings. If you do not intend to utilise this data, why was it included in the figures and diagrams?

Finally, please enhance your discussion by interpreting your data through your theoretical frameworks. This aspect is currently lacking in the discussion section.

Comments on the Quality of English Language

Line 452 – problem-solving skills

Line 576- acids and bases are presented twice

Author Response

Dear Referee
I made the modifications and corrections you made regarding the article titled "Problem-based Learning in Turkey: A Systematic Literature Review of Research in Science Education" and showed the necessary corrections on the article. If there is a missing explanation or if any other changes or corrections are required, you can inform me.

Thank you

Reviewer 2 Report

Comments and Suggestions for Authors

This study aims to conduct a systematic literature review to provide an overview of key findings and trends in Problem-Based Learning (PBL) studies within the context of science education in Turkey. A descriptive content analysis was employed to examine articles and postgraduate theses conducted in Turkey between 2000 and December 2023. A total of 133 studies were included, comprising 37 articles and 96 postgraduate theses, using an intentional sampling method known as criterion sampling. The results revealed that PBL had a positive impact on 34 different skills and no impact on another 11 skills. The preferred research design in all reviewed studies was quasi-experimental design. Limited inclusion of senior students was observed in samples from various school levels, and researchers predominantly favored physics topics for their studies.

The article is original, and its contribution is significant to the journal and scientific education. From my perspective, two aspects could be improved:

• If editorial guidelines permit and the editor deems it appropriate, it might be interesting to include a list of analyzed works for the research. This contribution can help readers understand in detail the conclusions drawn from the study. This list could be included as an annex. • The theoretical basis on problem-based learning could be enhanced and expanded. This topic is broad, and it may be useful to include literature reviews and classroom research from other countries. There are also very relevant articles on this topic in this journal that could help improve this work. Some of these works would enrich the manuscript

  • Pozuelo-Muñoz, J.; Calvo-Zueco, E.; Sánchez-Sánchez, E.; Cascarosa-Salillas, E. Desarrollo de habilidades científicas a través del aprendizaje basado en problemas en Educación Secundaria. Educativo. Ciencia. 2023, 13, 1096. https://doi.org/10.3390/educsci13111096
  • Himes, diputado; Agujas, HA; Krupa, EE; Borden, ML; Eagle, JL Investigación basada en proyectos (PBI) global durante una pandemia: una nueva perspectiva de la ecología del aprendizaje. Educativo. Ciencia. 2023, 13, 1099. https://doi.org/10.3390/educsci13111099
  • Leite, L., Dourado, L. y Morgado, S. (2016). La educación científica a través del aprendizaje basado en problemas: una revisión de investigaciones centradas en los estudiantes. Aprendizaje basado en problemas, 125.
  • Leite, L., Dourado, L. y Morgado, S. (2015). WebQuests “Sostenibilidad en la Tierra”: ¿califican como actividades de aprendizaje basado en problemas?. Investigación en Educación Científica, 45, 149-170.

Author Response

Dear Referee
I made the modifications and corrections you made regarding the article titled "Problem-based Learning in Turkey: A Systematic Literature Review of Research in Science Education" and showed the necessary corrections to the article. If there is a missing explanation or if any other changes or corrections are required, you can inform me.

Thank you

Round 2

Reviewer 2 Report

Comments and Suggestions for Authors

Dear authors.

The manuscript has improved considerably. In my opinion it is appropriate to publish it without any modification.

All the best.